# Electrical noise spectroscopy of magnons in a quantum Hall ferromagnet

Ravi Kumar[1,7], Saurabh Kumar Srivastav [1,7], Ujjal Roy[1,7], Jinhong Park [2,3,7], Christian Spånslätt [4], K. Watanabe [5], T. Taniguchi [5], Yuval Gefen [6], Alexander D. Mirlin[2,3] & Anindya Das [1]✉

Collective spin-wave excitations, magnons, are promising quasi-particles for next-generation spintronics devices, including platforms for information transfer. In a quantum Hall ferromagnets, detection of these charge-neutral excitations relies on the conversion of magnons into electrical signals in the form of excess electrons and holes, but if the excess electron and holes are equal, detecting an electrical signal is challenging. In this work, we overcome this shortcoming by measuring the electrical noise generated by magnons. We use the symmetry-broken quantum Hall ferromagnet of the zeroth Landau level in graphene to launch magnons. Absorption of these magnons creates excess noise above the Zeeman energy and remains finite even when the average electrical signal is zero. Moreover, we formulate a theoretical model in which the noise is produced by equilibration between edge channels and propagating magnons. Our model also allows us to pinpoint the regime of ballistic magnon transport in our device.

The emergence of charge-neutral collective excitations presents a powerful platform for developing data processing as well as information transfer with small power consumption. Among these excitations, spin-wave excitations, or their quanta 'magnons', in magnetic materials are promising. An obviously important task is to develop new techniques for the detection of these charge-neutral quasi-particles. So far, various experimental tools, such as inelastic neutron scattering[1,2], inelastic tunneling spectroscopy[3,4], terahertz spectroscopy[5,6], microwave Brillouin light scattering[7,8], nitrogen-vacancy center[9,10], and superconducting qubits[11] have been used to detect magnons in bulk magnetic materials. However, their detection in device geometries, which is necessary for information processing applications, has remained challenging until very recently. In particular, it was demonstrated by ref. 12 that magnons can be converted into electrical signals in a quantum Hall ferromagnet (QHF) in graphene.

Graphene offers a very versatile platform for new kinds of electronic devices. When subjected to a perpendicular magnetic field,

graphene shows several unique quantum Hall (QH) phases, related to its peculiar sequence of Landau levels (LL), manifesting both spin and valley degrees of freedom[13–15]. In particular, the particle-hole symmetric zeroth LL (ZLL) has a rich variety of QHF phases[16–22]: When the ZLL is partially filled, Coulomb interactions break spin and valley symmetries, and for a quarter ($v = -1$) or three-quarters ($v = 1$) filling, the QH phases comprise ferromagnetic insulator bulks with spin-polarized edge states[23–26]. While the charge excitations in the bulk of these QHF insulators have a gap determined by the exchange energy ($E_X \sim \frac{e^2}{\epsilon \ell_B}$, where $e$, $\epsilon$, and $\ell_B$ are the elementary charge, dielectric constant, and the magnetic length), the spin-waves (magnons) have instead a gap determined by the Zeeman energy ($E_Z = g\mu_B B$, where $g$ is the Landé g-factor, and $\mu_B$ is the Bohr magneton)[27] and are in fact the lowest energy excitations of the system. However, magnons do not carry electrical charge, and therefore do not have a large impact on electrical transport, which in turn makes it a difficult task to detect them. There are a few reported attempts of generating and detecting

[1]Department of Physics, Indian Institute of Science, Bangalore 560012, India. [2]Institute for Quantum Materials and Technologies, Karlsruhe Institute of Technology, 76021 Karlsruhe, Germany. [3]Institut für Theorie der Kondensierten Materie, Karlsruhe Institute of Technology, 76128 Karlsruhe, Germany. [4]Department of Microtechnology and Nanoscience (MC2), Chalmers University of Technology, S-412 96 Göteborg, Sweden. [5]National Institute of Material Science, 1-1 Namiki, Tsukuba 305-0044, Japan. [6]Department of Condensed Matter Physics, Weizmann Institute of Science, Rehovot 76100, Israel. [7]These authors contributed equally: Ravi Kumar, Saurabh Kumar Srivastav, Ujjal Roy, Jinhong Park. ✉e-mail: anindya@iisc.ac.in

spin-wave excitations or magnons in graphene-based QHF devices[12,28–32]. While magnon generation in these phases is based on an out-of-equilibrium occupation of edge channels with opposite spin, the detection of the magnons relies on the absorption of magnons by edge modes in the vicinity of ohmic contacts. The absorption of magnons by the edge modes creates excess electrons or holes in different corners of the graphene devices, and the measured electrical signal depends on the relative difference between the electron and hole signal magnitudes, which, in turn, critically depend on the device geometry. One may, therefore, not be able to detect any electrical signal if both the excited electrons and hole signals are equal. Thus, an alternative technique, which does not rely on the difference between excess electron and hole signals, is necessary for sensitive detection of magnons.

In this work, we demonstrate that electrical noise spectroscopy of magnons is a powerful method that satisfies the detection sensitivity requirement. We first establish that our device hosts symmetry-broken robust QH phases and study the magnon transport when the bulk filling is kept at $\nu = 1$. In order to generate magnons, we inject an edge current through an ohmic contact. While the injected current only flows in the downstream direction (as dictated by the electron motion subject to an external magnetic field), we measure the non-local electrochemical potential of a floating ohmic contact placed upstream from the source contact. Whenever the bias voltage of the injection contact corresponds to an energy smaller than the Zeeman energy $E_Z$, no non-local signal is detectable. As the bias energy exceeds $E_Z$, we measure a finite non-local signal for negative bias voltages. By contrast, the non-local signal remains zero for the entire positive bias voltages, which may naively suggest that magnons are not generated in this bias regime. Next, we switch to measuring the electrical noise and show that, as expected, no noise is detected below $E_Z$. On the other hand, as soon as the bias energy exceeds $E_Z$, the noise increases for both signs of the bias voltage. We show that the noise contributions created due to magnon absorption at different corners in our devices are additive, even when the average electron and hole currents mutually cancel (which happens for positive bias voltages). This renders noise spectroscopy a highly sensitive tool for magnon detection. Finally, our theoretically calculated noise captures well the experimental data and further suggests that the detected noise is a result of an increase in the effective temperature of the system as a result of equilibration between edge channels and magnons.

## Results

### Device and experimental principle

Figure 1a shows the schematics of our device and measurement setup. The device consists of *hBN* encapsulated graphite-gated high-mobility single-layer graphene, fabricated by the standard dry transfer technique[33,34]. Device fabrication and characterization are detailed in the Supplementary Information (SI-S1). The QH response of the device at a magnetic field (*B*) of 1 T is shown in Fig. 1c, indicating robust QH plateaus and the inset depicts the activation gap at $\nu = 1$, which is estimated to be ~4K (see SI-S1). As seen in Fig. 1a, the device has left and right ground contacts, while the upper transverse contact is utilized to inject current for magnon generation. The lower transverse contact is used to detect the change in the chemical potential of the floating contact (FC) due to magnon absorption. The device's bulk is tuned to the $\nu = 1$ QHF state, allowing it to host magnons. Importantly, the local doping due to the attached metallic contacts increases the filling factor to $\nu = 2$ near these contacts; this is represented (shown only for the right side of the FC) by additional loop-shaped edge modes at each contact, and are referred to as the "inner edge". In contrast, the outer edge propagates between contacts, as shown in Fig. 1a. A noiseless current, $I_{dc} + dI$, comprising a dc and an ac component, is injected into the red-colored source contact in Fig. 1a. The injected current flows along the outer edge with up-spin polarization.

This current exits the sample at the right-most grounded contact. The current along the inner edge, which flows around the source contact, has a down-spin polarization, does not contribute to the electrical conductance in the circuit. The dc voltage drop at the source contact, $V_S = I_{dc} \times \frac{h}{e^2}$, is shown as the electrochemical potential $\mu$ in Fig. 1a. The corresponding ac voltage that drops at the source contact is $dV_L = dI \times \frac{h}{e^2}$.

Whenever $\mu$ exceeds the Zeeman energy $E_Z$, i.e., $|\mu| \geq E_Z$, the electrons flowing along the circulating inner edge can tunnel into the outer edge by flipping their spin via magnon emission near point '**A**', as shown in Fig. 1a. This process does not directly alter the electrical conductance since the tunneling current flows back and is absorbed by the same injection contact. The emitted magnons propagate through the bulk of the device and can be absorbed at the device corners ('**B**', '**C**', '**D**', '**E**' and '**F**') via tunneling of electrons from the outer edge to the inner edge through the reverse spin flipping process. However, only parts of the currents generated at the two corners '**B**' and '**D**' arrive at the FC and contribute to the fluctuations of the electrochemical potential $\delta\mu_{FC}$ of the FC. This is so since generated electron and hole excitations are separated at points '**B**' and '**D**' into two respective currents, only one of which flows towards the FC, as shown in Fig. 1a, b. The fluctuations $\delta\mu_{FC}$ or noise are measured in the lower transverse contact placed to the left of the FC on the lower edge, by using an LCR resonance circuit at a frequency of ~740 kHz, followed by an amplifier chain and a spectrum analyzer[35,36]. At zero bias, the measured noise predominantly arises from the equilibrium thermal noise, $S_V(I = 0) = 4k_B TR$. At finite bias above the Zeeman energy, due to magnon absorption, excess voltage noise will be generated and quantified as $\delta S_V = S_V(I) - S_V(I = 0)$. The $\delta S_V$ is converted to excess current noise by $\delta S_I = \delta S_V / R^2$, where, $R = \frac{h}{\nu e^2}$ is the resistance of the considered QH edge. Further details about noise detection are specified in the Method section and in SI-S8.

We also measure the average chemical potential of the FC ($dV_{NL}$) via the same transverse contacts with standard lock-in measurements. It should be noted that the magnon generation in Fig. 1a is shown only for negative bias voltage; for positive bias voltage, magnons are instead generated near point '**E**', as shown in Fig. 2b. We carried out measurements in two devices, where for the second device (bilayer graphene), the filling near the contacts was tuned by local gating, showing similar results (see SI-S6, S7).

### Magnon detection using non-local resistance and noise spectroscopy

Figure 1d shows a 2D color map of the differential resistance $R_L = dV_L/dI$ (with *L* denoting "local") measured in the injection contact as a function of the bias voltage ($V_S$) and gate voltage ($V_{BG}$) around the center of the $\nu = 1$ plateau. It can be seen that within $E_Z$ [white vertical dashed lines in Fig. 1d], $R_L$ remains constant at $\frac{h}{e^2} \sim 25.8 k\Omega$ and decreases on both sides above $E_Z$, as shown by the solid magenta line in Fig. 1(d). This feature is similar to that in ref. 12, and can be understood as follows: For negative bias voltages, magnons are generated at '**A**'. Absorption at '**B**' and '**F**' reduce (via holes) and increase (via electrons) the chemical potential ($dV_L$) of the source contact, respectively, and thus affect $R_L$. However, since the absorption at '**B**' dominates over that at '**F**', $R_L$ decreases. Note that to be absorbed at '**F**', the magnons have to bend around the injected contact in contrast to their straight propagation when reaching '**B**'. Similarly, for positive bias voltage, the generated magnons from '**E**' [see Fig. 2b] are absorbed dominantly at '**F**' in comparison to '**B**' and thus $R_L$ decreases.

A more powerful approach to magnon detection, which permits to explicitly demonstrate and to explore magnon transport through the system, is provided by non-local measurements[12,31]. Figure 1e shows a 2D color map of the non-local differential resistance, $R_{NL} = dV_{NL}/dI$, vs bias and gate voltages, where $dV_{NL}$ is the chemical potential of the FC.

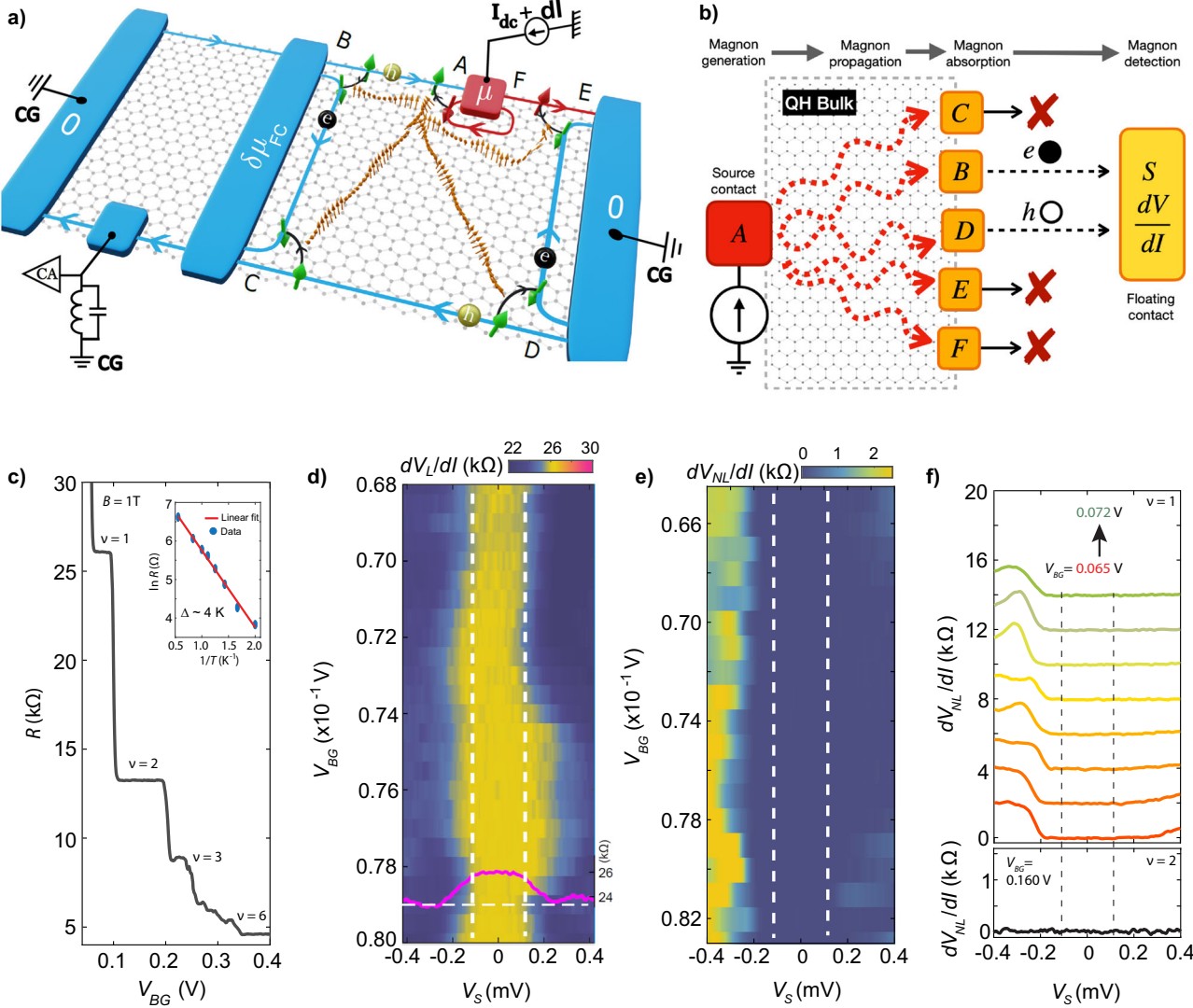

**Fig. 1 | Device schematic, magnon generation, and detection in quantum Hall ferromagnet. a** The device has a left, right, transverse, and floating contact. The device is set to $\nu = 1$, whereas regions adjacent to the contacts are tuned to $\nu = 2$, as shown by the additional circulating inner edges near the contacts. The spin polarization of the outer and inner edges are orthogonal, denoted by up and down arrows, respectively. A dc plus ac current ($I_{dc} + dI$) is injected through the upper red transverse contact, and when the electrochemical potential ($\mu$) exceeds Zeeman energy ($E_Z$), magnons are generated near point "**A**" via a spin-flip process. These magnons propagate through the QH bulk and are absorbed at other corners via the reverse spin-flip process. The bottom transverse contact is used to measure the voltage ($dV$) and noise ($S_V$) of the floating contact using standard lock-in ($\sim$13Hz)

and LCR resonance circuit ($\sim$740 kHz), respectively. **b** Magnon absorption at the different corners creates electron-hole excitations, but only points "**B**" and "**D**" contribute excess electrons and holes to the floating contact, respectively. **c** QH response at $B = 1$ T. The inset shows the activation gap of $\nu = 1$, which is $\sim$4 K. **d** 2D color map of the differential resistance ($dV_L/dI$) measured at the source contact vs the dc bias voltage ($V_S = I_{dc} \times \frac{h}{e^2}$) and the gate voltage around the center of the the $\nu = 1$ plateau. A line cut at $V_{BG} = 0.079$ is shown in solid magenta. **e** Non-local $dV_{NL}/dI$ of the floating contact vs source and gate voltages. **f** (upper panel) Line cuts from (**e**). Each plot is shifted vertically for clarity. (bottom panel) Non-local $dV_{NL}/dI$ for bulk $\nu = 2$. The vertical lines in **d**–**f** represent the Zeeman energy at B = 1 T.

As seen from the line cut in Fig. 1f (top panel), $R_{NL}$ remains zero within $E_Z$ (vertical, dashed lines), and increases for negative bias voltage above $E_Z$. However, $R_{NL}$ is almost zero for the entire positive bias voltage range. When the bulk filling was set to $\nu = 2$, no detectable non-local signal (Fig. 1f, lower panel) was observed as the ground state is then non-magnetic. The $R_{NL}$ in Fig. 1e, f can be understood as follows: As schematically shown in Fig. 1b, the magnon absorption at '**B**' and '**D**' contributes to the non-local signal of the FC via excess electrons and holes, respectively. For negative bias voltage, the magnons are generated at '**A**', but the absorption at '**B**' dominates over '**D**' due to shorter distance [Fig. 1a], and thus $R_{NL}$ takes a finite value. However, for positive bias voltage, the magnons are generated at '**E**' [Fig. 2b], and the absorption at '**B**' and '**D**' are almost equal due to their similar distance from '**E**'. Thus, $R_{NL}$ becomes almost zero.

Figure 2c shows a 2D color map of the measured excess noise ($S_I$) in the FC as a function of bias and gate voltages. The corresponding line cuts are shown in Fig. 2d (upper panel). We see that $S_I$ remains zero as long as $|eV_S| \leq E_Z$, and keeps increasing for larger values of either positive or negative bias voltage. This feature stands in stark contrast to Fig. 1e, f. We have repeated this measurement at different magnetic fields (see SI, S2). For example, Fig. 2e shows $R_{NL}$ and $S_I$ at $B = 2$ T, which display features very similar to the data at $B = 1$ T. The noise generation mechanism can be understood as follows: The absorption of magnons results in a change in the electrochemical potential of the FC either via excess electrons or holes, which are created at different absorbing corners. This process of magnon absorption at different corners occurs randomly, rendering the absorption events uncorrelated. When an equal number of excess electrons and holes reach the FC, the mean

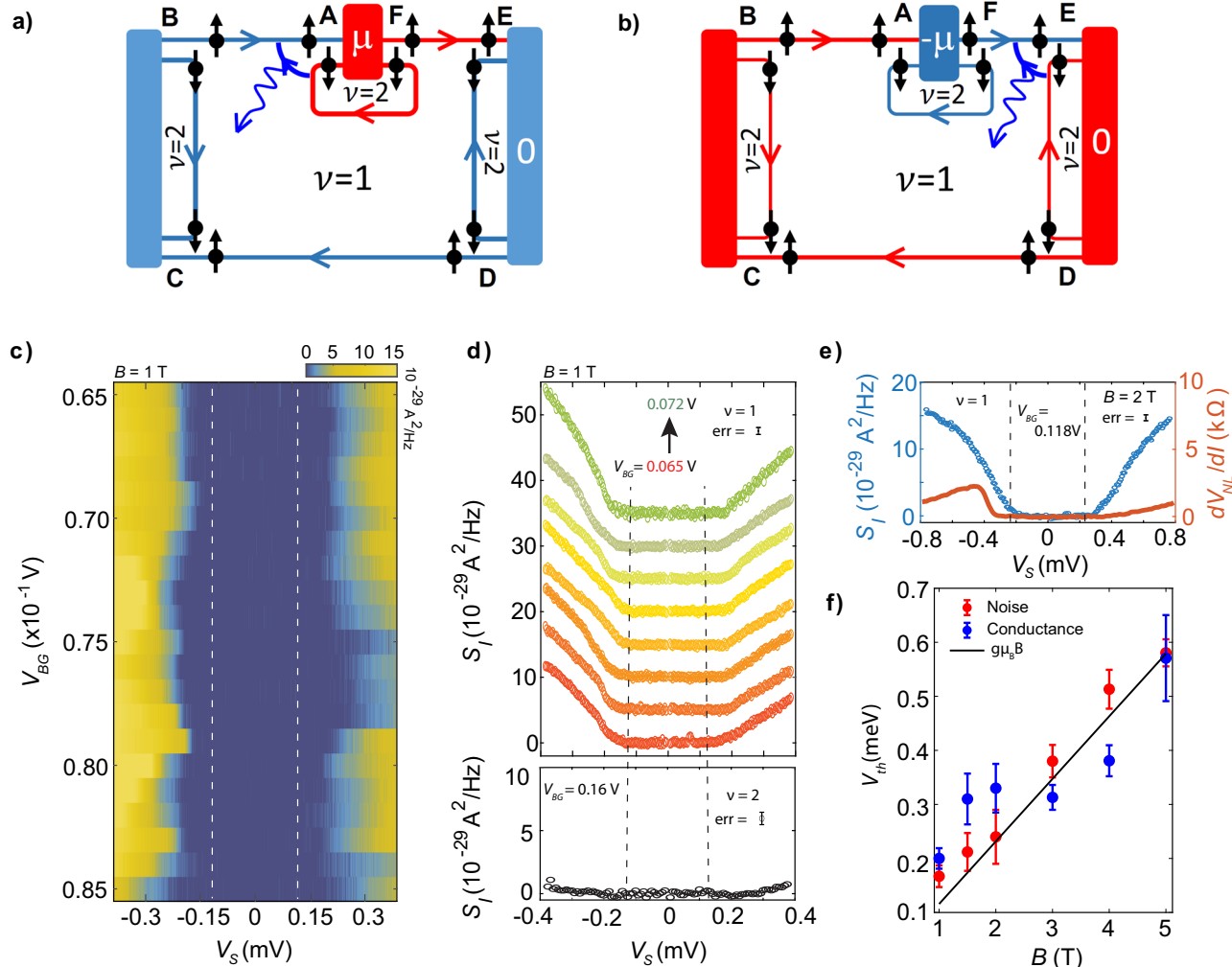

**Fig. 2 | Noise spectroscopy of magnons.** Magnon generation for negative (**a**) and positive (**b**) bias voltages, where magnons are generated at point **'A'** and **'E'**, respectively. The generated magnons propagate through the QH bulk and are absorbed at different corners. Only magnon absorption at points **'B'** and **'D'** generates noise at the floating contact. **c** 2D color map of excess noise generated at the floating contact for different bias and gate voltages. **d** (top panel) Line cuts of excess noise from c) for different $V_{BG}$ around the center of the $\nu=1$ plateau. Each plot is shifted vertically for clarity. (bottom panel) Noise spectra for bulk $\nu=2$. As expected, no excess noise is visible. **e** Noise spectra and $dV_{NL}/dI$ for bulk filling $\nu=1$ at $B=2$ T. The vertically dashed lines in **c**–**e** depict the Zeeman energy $E_Z$. Each data set in (**d**, **e**) is the five-point average of the raw data (shown in SI-Fig. 3), and "err" represents the standard deviation of the raw data. The "err" remains almost similar in magnitude for each data set in (**d**), and is shown for one of them. **f** Thresholds of the bias voltage (with error bars) vs magnetic field. The thresholds are extracted both from noise spectroscopy (solid red circles) and non-local differential resistance measurements (solid blue circles). Plotted is also $E_Z=g\mu_B B$ (solid black line).

electrochemical potential change of the FC is zero, resulting in a vanishing signal for the non-local resistance. In contrast, the variance of the electrochemical potential is independent of the signs of the impinging charges and thus remains nonzero because of the random arrival of excess electrons and holes, leading to fluctuations in the electrochemical potential of the FC.

Note that, in general, the finite noise measured in Fig. 2d may arise from hot phonons excited from Joule heating at the hot spot corners ('A' and 'E') in Fig. 2a and thus increase the effective temperature of the FC. In order to distinguish this noise from noise generated by magnon absorption, we study the non-magnetic state $\nu=2$ (which supports phonons but not magnons). No significant noise was detected for this non-magnetic state at $B=1$ T, as shown in Fig. 2d (bottom panel), and further shown in SI-Fig. 5 for higher magnetic fields. These results establish that the phonon contribution is negligible at magnetic fields $B \leq 2$ T.

The threshold voltage, $V_{th}$ for magnon detection, extracted from $S_I$ at different $B$, is plotted in Fig. 2f (solid red circles with error bars). At a given $B$, we calculate the root mean square (rms) value of the data,

and a sudden change in its magnitude is marked as the threshold voltage. The threshold voltage was extracted for several back gate voltage points across the Hall plateau, and its mean value and standard deviation as an error bar are shown in Fig. 2f. The detailed procedure of our $V_{th}$ extraction is discussed in SI–S3 and shown in SI-Fig 3. The solid black line in Fig. 2f represents the Zeeman energy $E_Z=g\mu_B B$ and is seen to closely follow the $V_{th}$ extracted from the noise data. We also show the threshold voltage extracted from the non-local resistance (for negative bias voltage) as solid blue circles with error bars. It can be seen from Fig. 2f that $V_{th}$ extracted from the non-local resistance exhibits a non-monotonic behavior with increasing $B$. This feature highlights the noise as a universal and robust probe for detecting magnons in contrast to the non-local resistance.

Note that the threshold voltage, $V_{th}$, above which the non-local resistance arises is significantly higher than $E_Z$ [see Fig. 2e, f] for $B < 3$ T. This behavior has been observed in previous works as well[12,32]. In contrast to the resistance data, however, the noise starts to increase at bias voltage $|eV_S| \sim E_Z$ [see Fig. 2d–f]. The difference in threshold voltages for the non-local resistance and the noise can be understood if

magnons are absorbed in **'B'** and **'D'** with equal probabilities within the bias voltage window $E_Z < |eV_S| < eV_{th}$. Hence, this absorption process is invisible in the non-local resistance data while strikingly visible in the noise data. Such an equal magnon absorption at **'B'** and **'D'** (geometrically located at asymmetric distances from the magnon generation point) for negative bias voltage may arise from ballistic magnon transport in the bias voltage window $E_Z < |eV_S| < eV_{th}$, where generated magnons propagate with a long wavelength $\lambda \gg \ell_B$. Such magnons experience little scattering from other degrees of freedom, particularly phonons or skyrmions[29], and reach all absorption corners with almost equal probabilities.

However, the ballistic motion of magnons may not be possible at a higher magnetic field, $B > 2$T. At higher $B$, a larger current is required to generate magnons, and thus also, more phonons may be excited at the hot spots near **'A'** and **'E'** [see Fig. 2a] due to increased Joule heating. Indeed, the proliferation of phonons is observed while measuring a finite noise for the non-magnetic state, $\nu = 2$, at $B > 2$T; see SI-Fig. 5. These excited phonons at higher $B$ can play an important role in scattering the magnons. As a result, the magnon transport may not remain ballistic, and hence the threshold voltage for the non-local resistance at $B > 2$T is reduced to the vicinity of $E_Z$, and in fact, is even slightly lower than $E_Z$ as seen in Fig. 2f. The reduction below $E_Z$ could be due to the temperature-broadening effect as the excited phonons elevate the temperature of the entire system and thus soften the Zeeman gap $E_Z$. In order to validate the claim about the temperature-broadening effect, we have measured the non-local resistance at $B = 1$T (where there are no phonons generated at the hot spot) at increasing bath temperature. We see the evolution of $V_{th}$ from higher than $E_Z$ at lower bath temperature to lower than $E_Z$ at higher temperature. These results are summarized in SI–S4. It is worth noting that while hot phonons contribute to the measured excess noise at higher magnetic fields ($B > 2$T), a distinct sudden increase in noise magnitude due to magnons occurs around $V_S \sim E_Z$, as shown in SI-Fig. 2.

## Theoretical model and comparison to experiment

In this section, we theoretically model the noise spectroscopy observed at a lower magnetic field, such that the effects of hot phonons can be neglected. We model the edge segments where the magnon generation and absorption take place as line junctions of co-propagating edges with length $L$, where electrons tunnel between edge channels (with spin-↑ and spin-↓), see Fig. 3a. Each such tunneling event is associated with the generation or absorption of magnons. We identify two distinct transport regimes depending on a degree of equilibration, characterized by the equilibration length $\ell_{eq}$; a short-junction regime ($L < \ell_{eq}$) with partial equilibration of the edge channels and the magnons, and a long-junction regime ($L > \ell_{eq}$) with strong equilibration, see Methods and SI-S9 for details. In the strong equilibration regime, equilibration in the magnon-generation region **'A'** takes place until the chemical potential difference between the edge channels equals $E_Z$. At this saturation point, further magnon generation is strongly suppressed. All generated magnons propagate in the bulk of the QH state and are eventually absorbed in one of the absorption regions (**'B'**, **'C'**, **'D'**, **'E'** and **'F'**). Each absorption event creates an electron-hole pair (an electron in the spin-↓ channel and a hole in the spin-↑ channel). These pairs produce the measured excess noise. In each absorption line junction, the excess noise is dominantly generated near $x = L$ [yellow circle in Fig. 3a] while remaining contributions are exponentially suppressed, see refs. 36–40 for a similar noise-generating mechanism. The excess noise $S_I$ reflects an increased temperature of the edge channels during the magnon absorption, given by

$$S_I = \frac{1}{2}\left(\frac{e^2}{h}(T_0 + T) - 2\frac{e^2}{h}T_0\right) = \frac{1}{2}\left(\frac{e^2}{h}(T - T_0)\right), \quad (1)$$

where $T_0$ is the bath temperature and

$$T = \sqrt{T_0^2 + \frac{3(|eV_S| - E_Z)(2E_Z + 3|eV_S|)}{(5\pi)^2}\theta(|eV_S| - E_Z)} \quad (2)$$

is the effective temperature of the system as a result of equilibration. Furthermore, $\theta(|eV_S| - E_Z)$ is the step function, which reflects the fact that no magnons can be absorbed for bias energies below $E_Z$. The factor 1/2 in Eq. (1) originates from the noise-measurement scheme, see Methods. In Fig. 3b, we compare our theoretically calculated excess noise (solid red line), $S_I$, with the experimentally measured noise versus the bias energy $eV_S$ (for simplicity, only the negative bias side is displayed), at fixed $T_0 = 20$ mK. Figure 3c shows the measured noise at different bath temperatures ($T_0$), and the corresponding theoretical plots are shown in Fig. 3d. A comparison between the experiment (orange circles) and theory (blue circles) for $S_I$ at $V_S = -0.3$ mV as a function of $T_0$ is shown in Fig. 3e. Our theoretical model captures well the characteristic features of the noise. Note that as seen in Fig. 3f, no excess noise was detected even at higher temperatures (600 mK) for $\nu = 2$ at $B = 1$ T.

The bias voltage dependence of the excess noise defines three regimes in Fig. 3b; (i) Biases $|eV_S| < E_Z$ result in no magnon generation and thus no excess noise. (ii) In a narrow region $0 < |eV_S| - E_Z < \frac{1}{\gamma L}$, the equilibration in magnon absorption and generation regions is only partial, $L < \ell_{eq} \equiv \frac{1}{\gamma(|eV_S| - E_Z)}$. Here, $\gamma$ is a parameter proportional to the tunneling strength in every tunnel junction comprising the line junction. This lack of equilibration allows us to model the magnon-generation and absorption regions as single tunnel junctions in regime (ii), see Methods and SI-S9 for further details of the model. In this model, the noise generation is of a non-equilibrium nature, resulting in $S_I = e^2 C(|eV_S| - E_Z)^2/h$ with the parameter $C = \gamma L$. The single parameter of the model, $C$, is obtained by fitting to the experimental data, as shown in Fig. 3b by the solid, blue line. (iii) For larger biases $|eV_S| > E_Z + \frac{1}{\gamma L}$ and hence $\ell_{eq} < L$, the edge channels and magnons achieve full equilibration in the magnon absorption and generation regions. We find that our theoretical model is in good agreement with the experimental data. In particular, at sufficiently large biases [regime (iii)], our equilibrated line junction model correctly describes several experimental observations: the sudden increase followed by (approximate) saturation of the non-local conductance as a function of the bias voltage [see Fig. 1f], the linear behavior of the noise as a function of the bias voltage [Fig. 2d], and the temperature dependence of the excess noise [Fig. 3d, e]. In addition, our single tunnel junction model [partial equilibration regime (ii)] properly describes the crossover region of bias voltages close to $E_Z$. Note that our theory assumes that magnons are absorbed in all the absorption regions with the same probability, but in reality, there may be deviations. These can explain some variations between experimental data curves.

## Discussion

As we show in Eq. (1), the excess noise generated in the line junction [regime (iii)] reflects the increase in temperature $T - T_0$ of the edge due to heating. The temperature behavior extracted from the measured excess noise data in Fig. 3b (right y-axis) is similar to the temperature behavior in Fig. 3d of ref. 32, a result which was obtained from $R_{xx}$ thermometry measurements. However, in the study by ref. 32, the equilibration between magnons with electrons or holes to form skyrmions is manifest in two distinct regimes. For shorter wavelengths ($\lambda \ll l_B$), equilibration occurs more easily than for longer wavelengths ($\lambda \gg \ell_B$). Consequently, a higher bias energy ($V_S > 4E_Z$) was necessary to induce the formation of skyrmions. The authors argued that a linear increment of the temperature with increasing $V_S$ originates from free magnons, whereas the saturation of temperatures at higher $V_S$ is attributed to the formation of skyrmions. Contrary to these findings, we have not encountered any

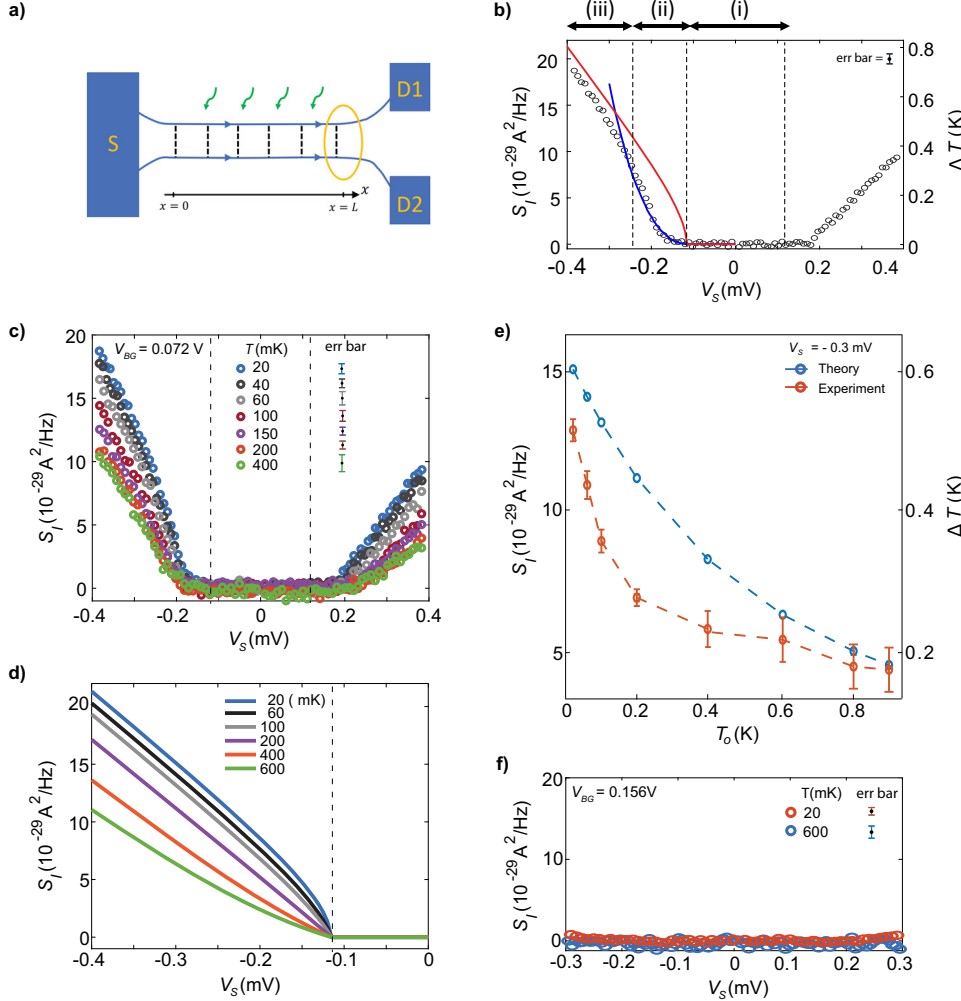

**Fig. 3 | Theoretical model and temperature dependence of excess noise. a** The magnon absorption (wiggly green lines with arrow) at any corner is modeled as a line segment of co-propagating edges, where tunneling of electrons occur from outer to inner edge [Figs. 1a, 2a, b]. The noise from the total tunneling current is dominantly generated in the vicinity of $x = L$ (yellow circle), where the local equilibrium noise dominates over the shot noise. **b** Comparison between the experimentally measured excess noise at 20 mK (black solid circles) and the theoretical results in a tunneling junction model; While the red solid line is the prediction for the strongly equilibrated regime, Eq. (1), the blue solid line is for the partially equilibrated regime which is obtained by fitting the formula $S_I = e^2\gamma L(eV_S - E_Z)^2/h$

with the experimental data. For this plot, we used the parameter choice $\gamma L = 0.8/E_Z$. These regimes of no magnons, partial and strong equilibration are further indicated by the horizontal arrows at the top of the axis. The right-hand side axis indicates the excess temperature, see Eq. (1). **c** Measured noise at different bath temperatures. (**d**) Noise calculated from Eqs. (1, 2) at different bath temperatures $T_0$. **e** Comparison between experiment and theory for the excess noise at $V_S = -0.3$ mV as a function of bath temperature. The right side of the axis indicates the excess temperature. **f** Excess noise for bulk filling $\nu = 2$ at 20 and 600 mK, both at $B = 1$T. In **b, c, f**, 'err bar' represents the standard deviation of the raw data shown in SI-Fig. 3.

temperature saturation at higher bias voltages in our experimental observations, indicating a lack of skyrmion formation. Therefore, we expect our experiment predominantly to examine free magnons. However, as previously described, their transport behavior can be hindered by hot phonons proliferated at higher $B$. Note that other possibilities, such as impurities or contact transparency, may affect the magnon transport at higher $B$.

Furthermore, the noise behavior as a function of the bias voltage appears to be correlated with that of the visibility of the Mach-Zender interferometry measured in ref. 30. It would be interesting to make a detailed connection between those two different quantities. Finally, we emphasize that our measurements were performed for relatively small magnetic fields and lower ambient temperatures than in previous works[12,30,32]. These small quantities allow us to fully neglect the effect of phonons. We have observed a sizeable effect of phonons only for magnetic fields $B > 2$T (see SI-S5). Finally, we expect that edge reconstruction does not happen in our graphite-gated devices, as theoretically studied in ref. 41 and experimentally established in

refs. 39,40,42,43. Even if edge reconstruction produces additional pairs of counter-propagating edge modes, it is possible that each individual pair localizes over a short-length scale. Thus, such modes do not contribute to the low-energy transport, yielding no quantitative changes in $R_{NL}$ and $S_I$.

In summary, we have demonstrated the utility of electrical noise spectroscopy as a highly sensitive tool for detecting and studying magnons in a quantum Hall ferromagnet. Our new protocol overcomes non-universal (e.g., device geometry dependent) features that screen out the presence of magnons, when other detection tools are employed, most prominently non-local conductance measurements. This robustness paves the way for utilizing magnons as low-power information carriers in future quantum technologies. Intriguing generalizations of our approach, with a promise of novel physics, include bulk phases of the fractional quantum Hall regime as well as of integer and fractional Chern insulator phases of twisted bilayer graphene[44-46]. Further implementations of our approach may include other ferromagnetic materials and vdW magnets[47,48].

## Methods

### Device and measurements scheme

Utilizing the dry transfer pick-up approach, we fabricated encapsulated devices consisting of a heterostructure involving hBN (hexagonal boron nitride), single-layer graphene (SLG), and graphite layers. The procedure for creating this heterostructure comprised the mechanical exfoliation of hBN and graphite crystals onto an oxidized silicon wafer through the widely employed scotch tape method. Initially, a layer of hBN, with a thickness of ∼25−30 nm, was picked up at a temperature of 90 °C. This was achieved using a poly-bisphenol-A-carbonate (PC) coated polydimethylsiloxane (PDMS) stamp on a glass slide attached to a home-built micromanipulator. The hBN flake was aligned over the previously exfoliated SLG layer picked up at 90 °C. The subsequent step involved picking up the bottom hBN layer of similar thickness. Following the same process, this bottom hBN was picked up utilizing the previously acquired hBN/SLG assembly. After this, the hBN/SLG/hBN heterostructure was employed to pick up the graphite flake. Ultimately, this resulting heterostructure (hBN/SLG/hBN/graphite) was placed on top of a 285-nm thick oxidized silicon wafer at a temperature of 180 °C. To remove the residues of PC, this final stack was cleaned in chloroform (CHCl3) overnight, followed by cleaning in acetone and isopropyl alcohol (IPA). After this, poly-methyl-methacrylate (PMMA) photoresist was coated on this heterostructure to define the contact regions using electron beam lithography (EBL). Apart from the conventional contacts, we defined a region of ∼6 μm² area in the middle of the SLG flake, which acts as a floating metallic reservoir upon edge contact metallization. After EBL, reactive ion etching (mixture of CHF₃ and O₂ gas with a flow rate of 40 sccm and 4 sccm, respectively, at 25 °C with RF power of 60 W) was used to define the edge contact. The etching time was optimized such that the bottom hBN did not etch completely to isolate the contacts from the bottom graphite flake, which was used as the back gate. Finally, the thermal deposition of Cr/Pd/Au (3/12/60 nm) was done in an evaporator chamber with a base pressure of ∼1 × 10⁻⁷ mbar. After deposition, a lift-off procedure was performed in hot acetone and IPA. The device's schematics and measurement setup are shown in Fig. 1a. The distance from the floating contact to the ground contacts was ∼5 μm, whereas the transverse contacts were placed at a distance of ∼2.5 μm.

All measurements were done in a cryo-free dilution refrigerator with a ∼20 mK base temperature. The electrical conductance was measured using the standard lock-in technique, whereas the noise was measured using an LCR resonant circuit at resonance frequency ∼740 kHz. The signal was amplified by a homemade preamplifier at 4 K, followed by a room temperature amplifier, and finally measured by a spectrum analyzer. At zero bias, the equilibrium voltage noise measured at the amplifier contact is given by

$$S_V = g^2(4k_B T R + V_n^2 + i_n^2 R^2) BW \,,\tag{3}$$

where $k_B$ is the Boltzmann constant, $T$ is the temperature, $R$ is the resistance of the QH state, $g$ is the gain of the amplifier chain, and BW is the bandwidth. The first term, $4k_B T R$, corresponds to the thermal noise, and $V_n^2$ and $i_n^2$ are the intrinsic voltage and current noise of the amplifier. At finite bias above the Zeeman energy, due to magnon absorption at points 'B' and 'D', chemical potential fluctuations of FC create excess voltage noise at the amplifier contact. At the same time, the intrinsic noise of the amplifier remains unchanged. Due to the white nature of the thermal noise and the excess noise, we could operate at a higher frequency (∼740 kHz), which eliminates the contribution from flicker noise (1/f) which usually becomes negligible for frequencies above a few tens of Hz. The excess noise ($\delta S_V$) due to bias current is obtained by subtracting the noise value at zero bias from the noise at finite bias, i.e., $\delta S_V = S_V(I) - S_V(I=0)$. The excess voltage noise $\delta S_V$ is converted to excess current noise $S_I$ according to $S_I = \frac{\delta S_V}{R^2}$, where $R = \frac{h}{\nu e^2}$ is the resistance of the considered QH edge.

### Theoretical calculation of the non-local resistance and noise

To compute the tunneling current, non-local resistance, and noise generated in the magnon absorption regions, we model the magnon generation and absorption regions as line junctions of length $L$. These line junctions are modeled as extended segments with two co-propagating edge channels in which electrons tunnel along a series of tunnel junctions, see Fig. 3a. We identify two distinct transport regimes: those of a short (partially equilibrated; $L < \ell_{eq}$) and long (equilibrated; $L > \ell_{eq}$) junctions, where $\ell_{eq}$ is the equilibration length. The short-junction regime can be equivalently modeled as a single tunnel junction. Details of the theoretical analysis are presented in SI, Sec. S9.

We first consider the partial-equilibration regime, treating the magnon generation or absorption regions as a single tunnel junction (at position $x = 0$). The Hamiltonian describing this junction reads

$$H = -i\upsilon \sum_{s=\uparrow,\downarrow} \int dx\, \psi_s^\dagger(x)\partial_x\psi_s(x) + \sum_q (E_Z + \hbar\omega_q)b_q^\dagger b_q$$
$$+ W\psi_\uparrow^\dagger(x=0)\psi_\downarrow(x=0)b^\dagger(x=0) + \text{h.c.}.\tag{4}$$

Employing the Keldysh non-equilibrium formalism, we derive zero-temperature expressions for the tunneling current $I_{ab}$, non-local resistance $dV_{ab}/dI$, and noise $S_{ab}$ generated in an absorption region, respectively:

$$I_{ab} = \gamma' \frac{e}{2h}(|eV_S| - E_Z)^2 \theta(|eV_S| - E_Z)\,,\tag{5}$$

$$\left|\frac{dV_{ab}}{dI}\right| = \gamma' \frac{h}{e^2}(|eV_S| - E_Z)\theta(|eV_S| - E_Z)\,,\tag{6}$$

$$S_{ab} = \frac{e^2}{h}\gamma'(|eV_S| - E_Z)^2\theta(|eV_S| - E_Z)\,.\tag{7}$$

Here, $\gamma'$ is a parameter associated with the tunneling strength in the tunnel junction. While the non-local resistance increases linearly with increasing bias voltage $eV_S$, the noise increases instead quadratically. For finite temperature, we first numerically determine the $eV_S$-dependence of the magnon chemical potential $\mu_m$, and thereby we obtain the $eV_S$-dependence of the non-local resistance and noise. This finite temperature result is used to fit the experimental data for regime (ii) in Fig. 3b.

In the limit of a long line junction, the last term in Eq. (4) is modified to describe tunneling in the spatial region $0 \le x \le L$. In the equilibrated regime, $L > \ell_{eq}$, this model yields non-local resistance and noise characteristics distinct from those in the single tunnel-junction model. Specifically, the equilibrated line-junction model predicts the following tunneling current, non-local resistance, and the excess noise in each individual absorption region,

$$I_{ab} = \frac{e}{2hM}(|eV_S| - E_Z)\theta(|eV_S| - E_Z)\,,\tag{8}$$

$$\left|\frac{dV_{ab}}{dI}\right| = \frac{h}{2Me^2}\theta(|eV_S| - E_Z)\,,\tag{9}$$

$$S_{ab} = \frac{e^2}{h}(T - T_0)\,,\tag{10}$$

with the increased temperature $T$ of the system, Eq. (2). Here, $M = 5$ is the number of absorption regions. Notably, the non-local resistance

(9) is constant in $eV_S$, whereas the noise [Eqs. (1)-(2)] instead increases linearly in $eV_S$ at sufficiently large bias voltage $eV_S$. In the calculation of Eqs. (2), (8), and (9), we have assumed for simplicity that the magnons are absorbed in each individual absorption region with equal probabilities. Note that the measured excess noise $S_I$ in Eq. (1) has the additional factor 1/2 compared with the excess noise generated in an absorption region, i.e., $S_I = \frac{1}{2} S_{ab} = 2 \times \frac{1}{4} S_{ab}$. The factor of 2 reflects contributions from two noise spots ('**B**' and '**D**') and the factor $1/4 = (1/2)^2$ originates from that only one channel out of the two emanating from the FC is measured at the bottom transverse contact, see Fig. 1a.

We also calculate the dependence of the equilibration length $\ell_{eq}$ on the bias voltage $eV_S$. We do this by using the results for the partial-equilibration regime (short $L$) and inspecting at what $L$ the equilibration becomes strong. The result reads

$$\ell_{eq} = \frac{1}{\gamma(|eV_S| - E_Z)}, \quad \text{for} |eV_S| > E_Z. \quad (11)$$

This equation implies a partial-equilibration regime for $|eV_S|$ slightly exceeding $E_Z$ and a strong-equilibration regime for larger $|eV_S|$, as discussed in the "Discussion" section above, and also illustrated in Fig. 3b. Equation (11) shows that $\ell_{eq}$ increases significantly as the bias energy approaches $E_Z$, indicating that the equilibration process takes place very slowly near $|eV_S| \sim E_Z$. This happens because the absorption rate per unit length is proportional to $(|eV_S| - E_Z)^2$ [Eq. (5)] whereas the total tunneling current in the equilibrated regime scales as $(|eV_S| - E_Z)$ [Eq. (8)].

## Reporting summary

Further information on research design is available in the Nature Portfolio Reporting Summary linked to this article.

## Data availability

The data presented in the manuscript are available from the corresponding author upon request.

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

## Acknowledgements

A.D. thanks the Department of Science and Technology (DST) and Science and Engineering Research Board (SERB), India, for financial support (SP/SERB-22-0387) and acknowledges the Swarnajayanti Fellowship of the DST/SJF/PSA-03/2018-19. A.D. also thanks financial support from CEFIPRA: SP/IFCP-22-0005. S.K.S. and R.K. acknowledge the Prime Minister's Research Fellowship (PMRF), Ministry of Education (MOE), and Inspire fellowship, DST for financial support, respectively. A.D.M., J.P., and Y.G. acknowledge support by the DFG Grant MI 658/10-2 and by the German-Israeli Foundation Grant I-1505-303.10/2019. Y.G. acknowledges support from the Helmholtz International Fellow Award, the DFG Grant RO 2247/11-1, CRC 183 (project C01), and the Minerva Foundation. Y.G. acknowledges the Infosys Chair professorship at IISc for this collaboration. C.S. acknowledges funding from the 2D TECH VINNOVA Competence Center (Ref. 2019-00068). This project has received funding from the European Union's Horizon 2020 research and innovation program under grant agreement No 101031655 (TEAPOT). K.W. and T.T. acknowledge support from the Elemental Strategy Initiative conducted by the MEXT, Japan, and the CREST (JPMJCR15F3), JST.

## Author contributions

R.K., S.K.S., and U.R. contributed to device fabrication, data acquisition, and analysis. A.D. contributed to conceiving the idea and designing the experiment, data interpretation, and analysis. K.W. and T.T. synthesized the hBN single crystals. J.P., C.S., Y.G., and A.D.M. contributed to the development of theory and data interpretation, and all the authors contributed to writing the manuscript.

## Competing interests

The authors declare no competing interests.
