## [Peer Review File · Nature Communications]

Reviewers' Comments:

Reviewer #1:

Remarks to the Author:

Magnons in quantum hall states is a very active field since [Wei et al. Science 362, 229 (2018)]. In this manuscript, the authors investigate the electrical noise properties induced by magnon emission in a graphene quantum Hall ferromagnet. The key point is that electrical noise enables magnon detection while the average electrical signal is zero.

The paper first verifies that magnons can be emitted and detected when the device is set to $\nu=1$. The non-local measurement can be explained with simple geometrical arguments. For negative bias, the emission point is located at site A and the two detection points are located at sites B and D. Since A is closer to B than to D, the non-local voltage is non zero. For positive bias, the emission point is located at site E and the two detection points are still located at sites B and D. Since the distance EB and ED are equal, the non-local voltage is zero.

The behavior of the electrical noise is different. In the autocorrelation shot noise, the fluctuations at each corners are additive, a non-zero noise is measured even for positive bias, confirming the presence of magnons.

A theoretical model is detailed to explain the different noise regimes. For bias smaller than the zeeman energy there is no magnon and no noise. In a first regime where the equilibration in magnon absorption and generation regions is only partial, $SI \propto V^2$ is found. For larger bias, measurements are compatible with a full equilibration configuration and $SI \propto V$.

The central idea of the paper is a bit technical and sample dependent. If the distance between the emission and detection site is slightly different a non-local voltage has to be measured. Concerning the proposed model, it can explain the noise behavior for negative bias, but for positive bias, the intermediate regime with $SI \propto V^2$ is missing. For these reasons, I do not feel that this study proposes key results that distinguish the manuscript from the existing references and can't recommend publications in Nature Communications.

Below are specific comments the authors could address:

_ When the emission point is located at site E, on the trajectory towards site B, there is a $\nu=2$ region where magnons can't propagate. This should lead to a finite non local voltage. Can the authors comment on this ?

_ Did the authors have the possibility to measure at larger magnetic field to recover the good linear dependence of the threshold as a function of the magnetic field ?

_ In Fig. 3b, the authors should add the error bars for the noise measurements. For positive voltage bias, $SI \propto V$ seems compatible with the measurements what corresponds to the Poissonian limit. Taking into account the error bars, could it be that $SI \propto V$ also for negative bias ?

_ Figure 3c-e-f error bars are missing.

Reviewer #2:

Remarks to the Author:

In this manuscript R. Kumar et. al. report on the combination of the electrical transport and noise measurements in the high-quality encapsulated single layer graphene sample with multiple electrical terminals. These contacts allow the authors to control magnon generation in the QH insulating regime when the device is tuned in to an overall $\nu=1$ QH state. By exploring the local and non-local transport of generated magnon, the authors generally resemble the previously reported data (e. g. in Ref. 12). Now, performing a noise spectroscopy, the authors report on the novel observations of the non-vanishing noise values for both positive and negative voltage bias.

At this point, the authors take a further look into the noise measurements by modeling the edge segment of their noise values where the absorption and generation takes place. By using short- and long-junction approximations (non-equilibrate and equilibrated edges) the authors seemingly find that in the low-bias regime the data may be described by the low-equilibrium model. In the high-bias regime, the edge equilibration is seemingly more stronger.

I find this work interesting. However, I would like to express a very cautious concern in the light of this observation and its following interpretation representing only an incremental advancement in the understanding or controlling magnon spin-waves in QH insulators. I believe that the manuscript in its current form will greatly benefit from addressing the following questions:

- 1) The key observation of the manuscript is a symmetrical shape of the noise magnon signal is additive for holes and electrons which is in a contrast to asymmetric transport data. The authors do not provide an expanded context as to why this is the case. I offer to provide more discussion about the noise detection mechanisms in the main text.
- 2) Paragraph 2 on page 7 discusses the difference in the threshold voltage for different bath temperatures and magnetic field values. In particular, the authors note that at high magnetic fields a ballistic propagation of magnons may be obscured by the scattering from phonon and skyrmions. I believe, here the abundance of proof is lacking. The authors only show 4 magnetic field points. What happens to the threshold at higher than 3T magnetic fields? I did not find any explanation about how the error bars were obtained. Fig. S2 does not seem to have much of significant threshold voltage difference between difference fields. In fact, it seems to be increasing for 3T. Fig. S3 further points at this as well.
- 3) Why would the author expect more scattering on the phonons in the presence of high magnetic fields? The authors reference the experimental work on skyrmion solids, where I can see a possible scattering that hinders and alternates the magnon transport, but the phonon argument seems to be more related to the temperature variation. The referenced work, however, observed the mentioned states in a much higher magnetic field (~ 10 T). Why would the authors expect such phases at 3 T in their work?
- 4) Some confusion arises after reading through the modeling of the reported data. In particular, I am concerned that the model used by the authors is not a parameter free. Coefficient γ used to obtain the theoretical curves is freely defined as a tunneling strength. As an experimentalist, I would appreciate a clearer context in the main text about the extensive modelling shown in the supplementary materials. Fig. S10 and 3d do not seem to reproduce the entirety of the features observed in the experiments while also trying to fit very subtle features where some significant dependence on the model parameters is expected. Please, provide a concise and clear context to the model in the main text.

Minor notes:

- 1) In the supplementary materials, I would suggest that the authors to include the actual optical micrograph of the studied samples if possible.
- 2) Fig. 2d lacks BG voltage labels for the linecuts. Please, add.

Taken together, I am not able to recommend the manuscript for publication in its current form, but I am open to review the revised version.

Reviewer #3:

Remarks to the Author:

The authors propose noise spectroscopy as a useful tool to detect magnons in graphene-based QH ferromagnet.

This is corroborated by noise measurements, together with non local resistance measurements, as a function of Voltage, showing that above E_Z magnons absorption processes are reflected in a finite noise signal, compatible with an effective temperature increase. This is consistent with a model that assumes equilibration process with electron tunneling between inner and outer edge channels, mediated by spin flip events.

The experimental results are clearly presented and, together with the supporting theoretical model, provide a very nice and convincing picture.

These results represent an important step forward in the understanding of neutral excitations and

may trigger new interests in the community.

I have only few suggestions for the authors:

- In the discussion section a more detailed comparison with previous works, such as the one by Pierce et al, would be important. In particular there, a relaxation mechanism (briefly mentioned here) between magnons and skyrmions has been discussed at length, while here it seems that this is not present.
- It is said that these measurements provide a way to assess ballistic magnon regime. However, this should be clarified and expanded in the main text.
- Could the authors comment on possible effects due to the presence of other edge channels or edge reconstruction mechanisms?

REPLIES TO REVIEWERS:

We hereby resubmit the manuscript NCOMMS-23-46681-T.

We express our gratitude to all Reviewers for thoroughly evaluating our work and providing constructive feedback. Their valuable insights have greatly contributed to enhancing the quality of our revised manuscript. Please find below our detailed response to all Reviewer's comments / questions / recommendations. While responding to each of the comments, we also specify in RED font our corresponding changes made in the Manuscript and Supplementary Information.

Reviewer #1 (Remarks to the authors)

Magnons in quantum hall states is a very active field since [Wei et al. Science 362, 229 (2018)]. In this manuscript, the authors investigate the electrical noise properties induced by magnon emission in a graphene quantum Hall ferromagnet. The key point is that electrical noise enables magnon detection while the average electrical signal is zero. The paper first verifies that magnons can be emitted and detected when the device is set to $\nu=1$. The non-local measurement can be explained with simple geometrical arguments.

For negative bias, the emission point is located at site A and the two detection points are located at sites B and D. Since A is closer to B than to D, the non-local voltage is non zero. For positive bias, the emission point is located at site E and the two detection points are still located at sites B and D. Since the distance EB and ED are equal, the non-local voltage is zero.

The behavior of the electrical noise is different. In the autocorrelation shot noise, the fluctuations at each corners are additive, a non-zero noise is measured even for positive bias, confirming the presence of magnons. A theoretical model is detailed to explain the different noise regimes. For bias smaller than the Zeeman energy there is no magnon and no noise. In a first regime where the equilibration in magnon absorption and generation regions is only partial, $SI \propto V^2$ is found. For larger bias, measurements are compatible with a full equilibration configuration and $SI \propto V$.

We thank the Reviewer for this concise summary of our results. Below is our detailed response to this Reviewer's comments.

Comment 1: The central idea of the paper is a bit technical and sample dependent. If the distance between the emission and detection site is slightly different a non-local voltage has to be measured.

Response: We thank the Reviewer for raising this comment, which, we understand, is a key criticism against publishing our manuscript. Designing a symmetric cancellation, or judging

whether indeed there is a symmetric cancellation in the nonlocal voltage, is very complicated and, in essence, impractical. A cancellation, or almost cancellation, may take place not only when the positions of two absorption points are symmetric; non-local voltage cancellation may happen even when the absorption points are positioned asymmetrically. One may clearly understand this feature by considering the regime of ballistic magnon transport: Scattering off gates or boundaries may render the magnon trajectories arriving at the two respective absorption points different, even if their distance from the magnon generation point is equal. Even when the magnon trajectories deviate from ballistic motion the resulting absorption at the respective points may be different. All this leads to the conclusion that (an almost full) cancellation of non-local voltage is a highly geometry and design-specific effect. Our reliance on noise measurements as a robust tool (independent of geometry and the nature of magnon transport) to detect magnons allows us to avoid this hindrance, and it is precisely the novelty of our approach.

This perspective is now emphasized on page 6 of the revised manuscript.

Comment2: Concerning the proposed model, it can explain the noise behavior for negative bias, but for positive bias, the intermediate regime with $S_I \propto V^2$ is missing. For these reasons, I do not feel that this study proposes key results that distinguish the manuscript from the existing references and can't recommend publications in Nature Communications.

Response: The Reviewer is certainly right in the observation that for a certain specific value of the back gate voltage, the noise signals measured for positive and negative bias may differ. This is now corroborated in Fig. 2d of the revised manuscript: the intermediate crossover behavior with $S_I \propto V^2$ is missing for the top most trace. However, considering Fig. 2d of the revised manuscript for other gate voltages, as well as SI-Figs. 2a,b,c of the revised manuscript, depicting other data sets of the magnetic field, the intermediate regime with $S_I \propto V^2$ is clearly visible. This statement applies to both signs of the bias voltages, with the exception of few curves. Furthermore, re-analyzing the data points of Fig. 2d and averaging them over different gate voltage values (equivalent to averaging over different positions on the QH plateau), we

recover the intermediate regime for positive bias, see attached Figure. This feature leads us to believe that small fluctuations of specific (non-averaged) data sets hinders observation of the intermediate behavior for positive bias voltages.

Considering the overall response depicted in Fig. 2d of the revised manuscript, the negative bias/positive bias asymmetry could be related to the asymmetric positioning of the various

elements of the circuit. For example, different magnon source points for positive (point E in Fig. 2b) and negative (point A in Fig. 2a) bias voltages, which may depend on the details of the local density variation near the Ohmic contact; the latter, in turn, may depend on the back-gate voltage.

Importantly, taking a broader perspective, notwithstanding certain asymmetries of the data curves, the “big universal message” of our study is that electrical noise facilitates magnon detection, even when a measurement of the average electrical input may give rise to (nearly) vanishing signals due to a non-universal device geometry and design.

Comment 3: When the emission point is located at site E, on the trajectory towards site B, there is a $\nu=2$ region where magnons can't propagate. This should lead to a finite non local voltage. Can the authors comment on this?

Response: We thank the Reviewer for asking this important question. Figure 1f and SI-Fig. 2a,b,c (right panels) show almost zero non-local voltage for positive bias. This applies to most values of the gate voltage except a few points. For ballistic magnon transport, one expects full cancellation irrespective of whether the paths are equivalent or not. Even when diffusive magnon transport is considered, the magnons can bend around the $\nu=2$ region and subsequently reach point B. Unless one works out details of the magnon propagation, and accounts for detailed geometrical components and possible coupling to external degrees of freedom, it is next to impossible to accurately determine whether we obtain full or partial cancellation of the non-local voltage. The main point of the current manuscript is that indeed the actual geometry or the trajectory of magnon propagation matters for non-local voltage, but our noise measurement is robust and universal and does not depend on particularities of the magnon propagation.

Comment 4: Did the authors have the possibility to measure at larger magnetic field to recover the good linear dependence of the threshold as a function of the magnetic field?

Response: We thank the Reviewer for asking this. As a response, we have now measured and included data at higher magnetic fields, 4T and 5T, depicted in SI-Fig.2 and SI-Fig.3 of the revised supplementary file. The additionally obtained threshold voltages from both non-local voltage and noise measurements are added in Fig. 2(f) of the revised manuscript. One can clearly see the linear dependence of the threshold voltage extracted from the noise as a function of the magnetic field following the Zeeman energy. By contrast, the threshold voltage extracted from the non-local voltage has a non-monotonic behaviour. This again establishes the robustness and universality of our noise measurements.

Comment 5: In Fig. 3b, the authors should add the error bars for the noise measurements. For positive voltage bias, $SI \propto V$ seems compatible with the measurements what corresponds to the Poissonian limit. Taking into account the error bars, could it be that $SI \propto V$ also for negative bias? Figure 3c-e-f error bars are missing.

Response: We thank the Reviewer for pointing out about the error bar. The data shown in Fig. 2d, Fig. 3b and 3c are 5-point average of the raw data. The representative raw data at different magnetic fields are shown in SI-Fig 3. We have estimated the error bar from the standard

deviation of each raw data set, and error bars are now shown in Fig. 2 and Fig. 3 of the revised manuscript. After including the error bars, it is very clear that for large (positive and negative) bias regime, $S_I \propto V$, but the intermediate crossover behavior ($S_I \propto V^2$) near the threshold for both bias voltages can be seen for most of the data set in Fig. 2 and Fig. 3.

Reviewer #2 (Remarks to the authors)

In this manuscript R. Kumar et. al. report on the combination of the electrical transport and noise measurements in the high-quality encapsulated single layer graphene sample with multiple electrical terminals. These contacts allow the authors to control magnon generation in the QH insulating regime when the device is tuned in to an overall $\nu=1$ QH state. By exploring the local and non-local transport of generated magnon, the authors generally resemble the previously reported data (e. g. in Ref. 12). Now, performing a noise spectroscopy, the authors report on the novel observations of the non-vanishing noise values for both positive and negative voltage bias.

At this point, the authors take a further look into the noise measurements by modeling the edge segment of their noise values where the absorption and generation takes place. By using short- and long-junction approximations (non-equilibrate and equilibrated edges) the authors seemingly find that in the low-bias regime the data may be described by the low-equilibrium model. In the high-bias regime, the edge equilibration is seemingly stronger.

I find this work interesting. However, I would like to express a very cautious concern in the light of this observation and its following interpretation representing only an incremental advancement in the understanding or controlling magnon spin-waves in QH insulators. I believe that the manuscript in its current form will greatly benefit from addressing the following questions:

We are delighted to read that Reviewer 2 found our work interesting. Our study reveals that the electrical noise resulting from magnon absorption is additive, even when average electron and hole currents cancel each other out. This renders noise spectroscopy a highly sensitive approach for magnon detection. Previous experiments in graphene-based QHF have examined their thermodynamic properties and excitonic nature through electrical conductance measurements. In this work, we introduce a new probe, noise spectroscopy, for detecting magnons, and show that noise is a universal and robust probe for detecting magnons in comparison to non-local resistance readings. The latter are undermined by non-universal features of the device, such as geometrical design. Further, our noise spectroscopy facilitates probing the magnon transport regime (whether diffusive or ballistic) as well as magnon-edge excitations equilibration characteristics, and pave the way to characterization of magnon-magnon interaction. In summary, our novel approach based on noise spectroscopy not only eliminates device geometry-dependent limitations that obscure the

detection of magnons (as seen in the non-local conductance measurements) but also opens the door to potential applications of our method in exploring new physics.

Comment 1: The key observation of the manuscript is a symmetrical shape of the noise magnon signal is additive for holes and electrons which is in a contrast to asymmetric transport data. The authors do not provide an expanded context as to why this is the case.

Response: We thank the Reviewer for asking this. The absorption of magnons results in a modification of the electrochemical potential of the floating contact (FC) via either excess electrons or holes, which are created at different absorbing corners. This process of magnon absorption at different corners occurs randomly, rendering the absorption events uncorrelated. The primary distinctions between non-local resistance and noise measurements are as follows:

- (i) In our non-local resistance measurements, we observe the alteration in the average chemical potential of the FC, while
- (ii) The noise measurements give access to the variance of the electrochemical potential of the FC.

When an equal number of excess electrons and holes reach the detector due to symmetrically positioned absorption sites relative to the magnon source, the mean electrochemical potential change of the FC is zero, resulting in a vanishing signal for the non-local resistance. However, the variance of the electrochemical potential remains nonzero because of the random arrival of excess electrons and holes, leading to fluctuations in the electrochemical potential of the detector. The potential variance is insensitive to whether it is electrons or holes (sign of the charges) that impinge on the FC. The potential average is highly sensitive, since the added electrons and holes shift the average potential in opposite directions. This is why the noise is a more sensitive probe for detecting magnons. In cases where the absorption points are asymmetrically positioned concerning the magnon generation point, there will be a finite change in the mean chemical potential, yielding a non-local resistance signal. However, the fluctuations in the chemical potential, which contribute to noise, largely remain unchanged. In our proposed geometry, as detailed in the manuscript, positive and negative bias voltages correspond to symmetric and asymmetric scenarios, respectively. Thus, non-local resistance remains sensitive to asymmetric configurations, while noise predominantly reflects symmetric conditions.

We have now emphasized these points in the revised manuscript (2nd paragraph of page 6) with the following elaboration:

“The noise generation mechanism can be understood as follows: The absorption of magnons results in a change in the electrochemical potential of the FC either via excess electrons or holes, which are created at different absorbing corners. This process of magnon absorption at different corners occurs randomly, rendering the absorption events uncorrelated. When an equal number of excess electrons and holes reach the FC, the mean electrochemical potential change of the FC is zero, resulting in a vanishing signal for the non-local resistance. In contrast, the variance of the electrochemical potential is independent of the signs of the impinging

charges and thus remains nonzero because of the random arrival of excess electrons and holes, leading to fluctuations in the electrochemical potential of the FC.”

Comment 2: I offer to provide more discussion about the noise detection mechanisms in the main text.

Response: Noise detection due to magnon absorption can be simply understood as following: At zero bias, we measure noise predominantly arising from a thermal noise contribution ($S_V(I=0) = 4k_B TR$). At finite bias above the Zeeman energy, due to magnon absorption, excess voltage noise will be generated. The excess noise, δS_V , due to the bias current is quantified by subtracting the noise value at zero bias from the noise at finite bias, i.e. $\delta S_V = S_V(I) - S_V(I=0)$. The excess voltage noise δS_V is then converted to excess current noise δS_I according to $\delta S_I = \delta S_V / R^2$, where $R = h/\nu e^2$ is the resistance of the quantum Hall edge. Note that conversion of δS_V to δS_I is simply a matter of convenience and is mainly used to compare with the theoretical results. Details about the noise detection mechanism is discussed further in the method section of the manuscript.

We have also added the following sentences in the revised manuscript (1st paragraph of page 5).

“At zero bias, the measured noise predominantly arises from the equilibrium thermal noise, $S_V(I=0) = 4k_B TR$. At finite bias above the Zeeman energy, due to magnon absorption, excess voltage noise will be generated and quantified as $\delta S_V = S_V(I) - S_V(I=0)$. The δS_V is converted to excess current noise by $\delta S_I = \delta S_V / R^2$, where, $R = \frac{h}{\nu e^2}$ is the resistance of the considered QH edge. Further details about noise detection are specified in the Method section and in SI-S8.”

Comment 3: Paragraph 2 on page 7 discusses the difference in the threshold voltage for different bath temperatures and magnetic field values. In particular, the authors note that at high magnetic fields a ballistic propagation of magnons may be obscured by the scattering from phonon and skyrmions. I believe, here the abundance of proof is lacking. The authors only show 4 magnetic field points. What happens to the threshold at higher than 3T magnetic fields?

Response: We thank the Reviewer for asking this question. Since our initial submission, we have conducted measurements at higher magnetic fields, specifically 4T and 5T, which are now illustrated in the revised supplementary figures (Supplementary Fig. 2, Fig. 3, and Fig. 5). Additionally, we have updated the dependence of threshold voltages on magnetic field in Fig. 2f of the revised manuscript, and also added the figure below for the Reviewers convenience.

This figure indicates that while the threshold bias voltage extracted from the noise data follows well the line $|eV_S| \sim E_Z$, the threshold bias voltage extracted from the non-local conductance data exhibits a non-monotonic trend. For magnetic fields $B < 3\text{T}$, within the bias range $E_Z < |eV_S| < eV_{th}$, the threshold bias voltage from the non-local conductance is notably higher than the Zeeman energy E_Z . However, for $B \geq 3\text{T}$, it diminishes to become comparable to or smaller than E_Z .

This discrepancy in threshold voltages between non-local resistance and noise data can be explained as follows: If magnons are absorbed at points B and D (Fig. 1a of the revised manuscript) within the bias voltage window of $E_Z < |eV_S| < eV_{th}$ with equal probabilities, this absorption process remains unseen in the non-local resistance data but is distinctly visible in the noise data. Such equal magnon absorption may stem from ballistic magnon transport within the bias voltage window $E_Z < |eV_S| < eV_{th}$. Here, the generated magnons propagate with a long wavelength ($\lambda \gg \ell_B$), experiencing minimal scattering off other degrees of freedom, particularly phonons or skyrmions. However, if the magnon transport is not ballistic, then equal magnon absorption rates is more delicate, and hence, non-local resistance is expected to give a finite signal only for $|eV_S| \geq E_Z$.

At higher magnetic fields, ballistic magnon transport may not be feasible due to the need for a higher bias voltage (since the Zeeman energy becomes higher) to generate magnons. This results in stronger hotspots due to Joule heating, which easily excite phonons, subsequently scattering magnons and hindering their ballistic motion. Consequently, the assumption of equal probability of magnon absorption at B and D no longer holds, leading to a reduced threshold voltage for the non-local conductance at higher magnetic fields, $|eV_{th}| \sim E_Z$, possibly even slightly lower than E_Z due to temperature broadening effects, which is described in SI-section S4.

Confirmation of phonon generation can be observed by measuring the noise in a non-magnetic state ($\nu = 2$). Although our experiment's geometry precludes direct electrical current flow from the current-injected source contact to the detector contact, the generation of hot phonons can increase the effective temperature of the detector, resulting in finite noise. Phonon effects are evident for $B > 2\text{T}$ in our noise measurements, even at bias voltages smaller than Zeeman energy (Supplementary Fig. 5).

We have produced the evidence of phonons at higher magnetic fields for three values (3T, 4T and 5T) and reemphasized this feature in the revised manuscript (page 8).

Though finite noise due to phonon at $B = 3T, 4T,$ and $5T$ was detected, but its magnitude remains smaller than the total noise measured for the $\nu = 1$ quantum Hall ferromagnetic phase within our working bias range, as shown in Supplementary Information. The sudden change in the slope of noise around E_z is evident due to magnon absorption.

Comment 4: I did not find any explanation about how the error bars were obtained.

Response: Revised supplementary Fig. 3 shows the way how the threshold voltages (V_{th}) from the non-local resistance and noise data were determined at different magnetic fields. We calculate the rms values of the data, and a sudden change in its magnitude is marked as a threshold voltage, which is shown by the vertical black lines in the revised supplementary Fig. 3. At a given magnetic field, the V_{th} was extracted for several back-gate voltage points across the plateau for the data set shown in revised supplementary Fig. 2, and its mean value and standard deviation (as error bar) are shown in Fig. 2(f) in the revised main manuscript. Note that for the noise data at 4T and 5T, V_{th} is calculated from the sudden change in the slope of noise as there is finite background noise at higher magnetic fields, coming from phonons as shown in supplementary Fig. 3.

We have added the following sentence in the revised manuscript (page 6).

“The threshold voltage, V_{th} for magnon detection, extracted from S_{I-I} at different B , is plotted in Fig. 2(f) (solid red circles with error bars). At a given B , we calculate the root mean square (rms) value of the data, and a sudden change in its magnitude is marked as the threshold voltage. The threshold voltage was extracted for several back gate voltage points across the Hall plateau, and its mean value and standard deviation as an error bar are shown in Fig. 2(f). The detailed procedure of our V_{th} extraction is discussed in SI-S3 and shown in SI-Fig 3. The solid black line in Fig. 2(f) represents the Zeeman energy $E_Z = g \mu_B B$ and is seen to closely follow the V_{th} extracted from the noise data. We also show the threshold voltage extracted from the non-local resistance (for negative bias voltage) as solid blue circles with error bars. It can be seen from Fig. 2(f) that V_{th} extracted from the non-local resistance exhibits a non-monotonic behaviour with increasing B . This feature highlights the noise as a universal and robust probe for detecting magnons in contrast to the non-local resistance.

Comment 5: Fig. S2 does not seem to have much of significant threshold voltage difference between difference fields. In fact, it seems to be increasing for 3T. Fig. S3 further points at this as well.

This is indeed the case, in stark distinction from threshold voltages extracted from the noise, which remain linear and follows $E_Z = g\mu_B B$ (Fig. 2f of the revised manuscript as well as the above Figure). By contrast, the threshold voltage extracted from the non-local resistance is highly non-monotonic. It remains significantly higher than E_Z and almost constant for 1.5, 2 and 3T. However, beyond 2T it starts increasing around the vicinity of E_Z . This non-monotonic response at smaller magnetic field ($B < 3T$) has been explained by ballistic magnon transport.

This feature has been reemphasized in page 6 and page 8 of the revised manuscript.

Comment 6: Why would the author expect more scattering on the phonons in the presence of high magnetic fields? The authors reference the experimental work on skyrmion solids, where I can see a possible scattering that hinders and alternates the magnon transport, but the phonon argument seems to be more related to the temperature variation. The referenced work, however, observed the mentioned states in a much higher magnetic field (~10 T). Why would the authors expect such phases at 3 T in their work?

Response: While answering comment 3 above, we have clarified why more phonons are expected to be excited at higher magnetic fields, and how these excited phonons can hinder the ballistic motion of magnons. Here, we briefly summarize this in the following paragraph, which is included in the revised manuscript.

At higher B, a larger current is required to generate magnons (since the EZ increases) and thus more phonons can be excited from the hotspots near 'A' and 'E' [see Fig. 1(a) in the revised manuscript] due to increased Joule heating. These excited phonons at higher B can play an important role in scattering the magnons; as a result, the magnon transport may not remain ballistic at higher B. The proliferation of phonons is experimentally observed while measuring a finite noise for the non-magnetic state, $\nu = 2$, at $B > 2T$; see SI-Fig. 5 in the revised supplementary information.

Regarding how Skyrmions are formed and interact with magnons – this is described by Pierce et al. (Nature Physics 18, 37–41 (2022)). In that work, the authors observed that when $V_S < 4E_z$, the generated magnons are free and travel through the Quantum Hall (QH) bulk, eventually reaching various absorbing corners. Conversely, when $V_S > 4E_z$, the density of magnons increases rapidly. Many of these magnons bind with an electron or a hole, forming skyrmions, which are bound states predominantly fixed at specific positions. These bound magnons do not significantly contribute to magnon transport.

The level of equilibration between free magnons and skyrmions depends on the wavelength of the magnon. For shorter wavelengths ($\lambda \ll \ell_B$), equilibration/thermalization is easier compared to longer wavelengths ($\lambda \gg \ell_B$). Hence, a higher bias energy ($V_S > 4E_z$) is required to excite short-wavelength magnons, which are observed to bind or thermalize with skyrmions. The distinction between free and bound magnons can be identified by measuring the change in effective temperature through variations in longitudinal resistance at magnon absorption points. For free magnons, the temperature increases almost linearly whereas it saturates when the skyrmions start forming.

In our experiment, the bias energy typically utilized is lower than the critical energy needed to induce the formation of skyrmions ($V_S > 4E_z$). As a result, we do not anticipate the generation of skyrmions. This is corroborated by the absence of temperature saturation (as inferred from excess noise) within our operational range of bias energy. We expect, therefore, our experiments predominantly examine free magnons. However, their transport can be hindered by phonons proliferated at $B > 2T$ as previously answered in comment 3.

We have included a detailed discussion in the revised manuscript (page 11).

Comment 7: Some confusion arises after reading through the modeling of the reported data. In particular, I am concerned that the model used by the authors is not a parameter free. Coefficient γ used to obtain the theoretical curves is freely defined as a tunneling strength. As an experimentalist, I would appreciate a clearer context in the main text about the extensive modelling shown in the supplementary materials. Fig. S10 and 3d do not seem to reproduce the entirety of the features observed in the experiments while also trying to fit very subtle features where some significant dependence on the model parameters is expected. Please, provide a concise and clear context to the model in the main text.

Response: Yes, as the Reviewer mentioned, the tunneling coefficient γ is a free parameter of the model. As specified in the main text, there are three transport regimes: (i) no magnons at $V_S < E_Z$, (ii) partial equilibration regime in the intermediate bias, and (iii) full equilibration regime at the high bias. Crucially, the regime (iii) is fully captured by the model with a full equilibration of magnons with edge channels. This regime does not have any free parameter, see Eq. (1) in the main text. On the other hand, the partial equilibration regime (ii) follows $S_I = e^2 C (eV_S - E_Z)^2 / h$, Eq. (7) in Method section, obtained from a tunneling-junctions model. Here, $C \equiv \gamma L$ is the only free parameter for regime (ii) and we obtain it by fitting with the experimental data. Overall, with the single parameter C , the entire curve S_I vs V_S (noise as a function of bias voltage) is quite nicely fitted as shown in Fig. 3b.

Motivated by this Reviewer question, in the revised version we have

(1) Changed the main text (page 10) to clarify the parameter of the model:

“In this model, the noise generation is of non-equilibrium nature, resulting in $S_I = e^2 C (eV_S - E_Z)^2 / h$ with the parameter $C = \gamma L$. The single parameter of the model, C , is obtained by fitting to the experimental data, as shown in Fig. 3(b) by the solid, blue line.

(2) In the main text (page 10), we have now referred to Method section where the theoretical model is presented for more details.

(3) We have now added the obtained fitting parameter in the figure caption of Fig. 3.

Comment 8: In the supplementary materials, I would suggest that the authors to include the actual optical micrograph of the studied samples if possible.

Response: We have now added the optical image of the device in Fig 1 of the revised supplementary.

Comment 9: Fig. 2d lacks BG voltage labels for the line cuts. Please, add. Taken together, I am not able to recommend the manuscript for publication in its current form, but I am open to review the revised version.

Response: We thank Reviewer for pointing out this. We have added the BG voltage labels in Fig 2d as well as in other figures of the revised manuscript.

Reviewer #3 (Remarks to the authors)

The authors propose noise spectroscopy as a useful tool to detect magnons in graphene-based QH ferromagnet. This is corroborated by noise measurements, together with non local resistance measurements, as a function of Voltage, showing that above E_Z magnons absorption processes are reflected in a finite noise signal, compatible with an effective temperature increase. This is consistent with a model that assumes equilibration process with electron tunneling between inner and outer edge channels, mediated by spin flip events.

The experimental results are clearly presented and, together with the supporting theoretical model, provide a very nice and convincing picture. These results represent an important step forward in the understanding of neutral excitations and may trigger new interests in the community. I have only few suggestions for the authors:

We thank the Reviewer for this concise summary of our work. We are very pleased to learn that the Reviewer finds our results an important step towards the understanding of the kinetics of neutral excitations. Furthermore, we are delighted to read that the Reviewer praises our manuscript for its clear presentation and significance.

Comment 1: In the discussion section a more detailed comparison with previous works, such as the one by Pierce et al, would be important. In particular there, a relaxation mechanism (briefly mentioned here) between magnons and skyrmions has been discussed at length, while here it seems that this is not present.

Response: We thank the reviewer for commenting on this. The work of Pierce et al. (Nature Physics 18, 37–41 (2022)) has investigated the thermodynamics of both free and bound magnons with bias energy. Their observations indicate that when $V_S < 4E_Z$, the generated magnons are free and travel through the Quantum Hall (QH) bulk, eventually reaching various absorbing corners. Conversely, when $V_S > 4E_Z$, the density of magnons increases rapidly. Many of these magnons bind with an electron or a hole, forming skyrmions, which are bound states predominantly fixed at specific positions. These bound magnons do not significantly contribute to magnon transport.

The level of equilibration between free magnons and skyrmions depends on the wavelength of the magnon. For shorter wavelengths ($\lambda \ll \ell_B$), thermalization is easier, contrasting this with longer wavelengths ($\lambda \gg \ell_B$). Hence, a higher bias energy ($V_S > 4E_Z$) is required to excite short-wavelength magnons, which are bound to skyrmions. The distinction between free and bound magnons can be identified by measuring the change in the effective temperature through variations in longitudinal resistance at magnon absorption points. For free magnons, the temperature increases with the bias voltages almost linearly, whereas it saturates when skyrmions begin to form. Additionally, compressibility measurements reveal an effective reduction in the QH bulk gap due to the formation of bound skyrmions.

In our experiment, the bias energy typically utilized is lower than the critical energy needed to induce the formation of skyrmions ($V_S > 4E_Z$). As a result, we do not anticipate the generation of skyrmions. This is corroborated by the absence of temperature saturation (as inferred from excess noise) within our operational range of bias energy. In summary, our experiment predominantly examines free magnons. Their transport is instead hindered by phonons proliferated at $B > 2T$ as specified in the main text.

We have now added a detailed discussion of this physics in the revised manuscript (page 11).

Comment 2: It is said that these measurements provide a way to assess ballistic magnon regime. However, this should be clarified and expanded in the main text.

Response: We thank the Reviewer for commenting on this. The relationship between the threshold voltage (obtained from both noise and non-local resistance measurements) and the magnetic field strength serves as a valuable means to evaluate the nature of magnon transport. In the revised manuscript, we have extended our measurements to include higher magnetic fields (3T and 4T), as illustrated in Fig. 2f. For the Referee's convenience, we have retained the relevant figure below.

The figure above illustrates that while the threshold bias voltage derived from noise data follows $|eVS| \sim EZ$, the threshold bias voltage obtained from non-local resistance data displays a non-monotonic behavior. For magnetic fields $B < 3T$, within the bias range $EZ < |eVS| < eV_{th}$, the threshold bias voltage for non-local resistance notably exceeds the Zeeman energy EZ . This higher threshold voltage ($> EZ$) observed in non-local resistance at lower magnetic fields suggests potential ballistic magnon transport.

In the context of ballistic magnon motion, even with magnon absorption points asymmetrically positioned from the magnon source point (negative bias voltage in our experiment), such as at points 'B' and 'D' (refer to Fig. 1a of the revised manuscript), within the bias voltage window $EZ < |eVS| < eV_{th}$, equal probabilities of magnons being absorbed at these points are anticipated. Consequently, this scenario results in a zero signal in non-local resistance data while demonstrating a finite signal in noise data.

This observation is unsurprising, as within the bias voltage window $EZ < |eVS| < eV_{th}$, the generated magnons propagate with long wavelengths ($\lambda \gg \ell_B$), experiencing minimal scattering from other degrees of freedom such as phonons or skyrmions. Consequently, these magnons can reach every absorption corner with nearly equal probability.

We have clarified and expanded in page 8 of the revised manuscript.

Comment 3: Could the authors comment on possible effects due to the presence of other edge channels or edge reconstruction mechanisms?

Response: This is indeed a very relevant question. The dynamics of reconstruction is intricately tied to factors such as Coulomb interactions at the edge and the confinement potential. In the context of graphene-based systems, a theoretical proposal [Zi-Xiang Hu et al, PRL 107, 236806 (2011)] suggests that in graphite gated devices with sharper confining potentials, edge reconstruction may not be favorable. This theoretical assertion, in agreement with general studies of edge reconstruction [Phys. Rev. Lett. 129, 146801 and Low Temp. Phys. 48, 420–427 (2022)], finds experimental support in studies such as those conducted via scanning tunneling microscopy (STM) (Nature Communications 4, 1744 (2013) & Sci. Adv. 9, eadf7220 (2023)), as well as in our prior investigations on thermal conductance measurements (Science Advances 5 (no 7, eaaw5798) & Nature Communications 13 (1), 5185).

Notwithstanding the above arguments against edge reconstruction, let us assume it does take place, and assess its potential impact on our measurements. The most probable manifestation of edge reconstruction at filling 1 has been outlined in Ref. Low Temp. Phys. 48, 420–427 (2022), suggesting the formation of one upstream and two downstream channels. Considering such a reconstructed structure, we address various scenarios of equilibration among the counter-propagating edge modes, and their impact on our schemes of conductance and noise measurements. To keep it simple, we will consider external bias whose energy scale is smaller than the Zeeman energy. With this, we exclude magnon generation due to the bias. We next consider the following scenarios:

(i) No equilibration (both electrically and thermally) – this would result in a finite electrical signal and noise at our detector (FC).

(ii) No thermal equilibration but full electrical equilibration (as observed in hole-like FQH states in graphene, Phys. Rev. Lett. 126, 216803 (2021) and Nature Communications 13, 213 (2022)) – this would yield no electrical signal but finite noise at the detector.

(iii) Full equilibration of both electrical and thermal – this would lead to neither electrical signal nor noise.

In our experimental procedure, both electrical signal and excess noise remain zero for bias range smaller than the Zeeman energy. These results rule out the first two scenarios, yet do not preclude the possibility of full equilibration. Under full equilibration, a single resultant downstream mode would effectively emerge, exhibiting electrical and thermal conductance akin to that of a bare single edge at filling 1. Consequently, we anticipate similar outcomes regarding the separation of excited electrons and holes due to magnon absorption, yielding similar results as for the no-reconstruction scenario.

We have added a discussion on this in the revised manuscript (last paragraph on page 11).

Reviewers' Comments:

Reviewer #1:

Remarks to the Author:

The revised version has incorporated the feedback from the reviewers, resulting in substantial improvements. I am pleased with the responses provided and recommend this manuscript for publication in Nature Communications.

Reviewer #2:

Remarks to the Author:

The revised version of the manuscript answers my concerns raised in the previous round of review. The rebuttal letter provides sufficient answers to my questions, clarifies importance of the noise measurements over transport and provides additional data that solidifies the discussion (e. g. higher magnetic field data). The current version of the manuscript and its main conclusions appear more solid.

I retain a small doubt about the explanation for the phonon generation at high magnetic field vs. skyrmions. There are two things that appear troublesome. 1) The authors do not consider other contributors to the high-field magnon transport such as impurities or contact transparency (STM experiments examined local spectra, while in the current manuscript magnons transport over a few micrometers across the sample). 2) The data in Fig. S5 show that the lower V_s/V_Z needed to initiate alleged scattering on phonon at higher magnetic fields. In particular, it seems that if I extrapolate the threshold from the $B=5T$ to $B=2T$ (taken that the Zeeman energy will change by a factor of 2.5), the actual bias voltage that is needed to create an imbalance between "B" and "D" is the same for all fields except 1.5 T. This does not provide sufficiently strong argument towards phonon-generated scattering in the light of point 1.

It does not undermine the integrity of the main conclusions of the manuscript since the majority of the data was taken at lower magnetic fields and lower bias voltages. But I would recommend that the authors tone down discussions in the main text and remain open for other possibilities in the actual quantum Hall device they study.

I can now recommend this manuscript for publication in Nature Communications.

Reviewer #3:

Remarks to the Author:

I have read the updated version of the manuscript and the answers to the Referees' criticisms. I think that the paper in the present version is suitable for publication.

Reviewer #1 (Remarks to the authors)

The revised version has incorporated the feedback from the reviewers, resulting in substantial improvements. I am pleased with the responses provided and recommend this manuscript for publication in Nature Communications.

We thank the Reviewer for accepting our manuscript for the publication.

Reviewer #2 (Remarks to the authors)

The revised version of the manuscript answers my concerns raised in the previous round of review. The rebuttal letter provides sufficient answers to my questions, clarifies importance of the noise measurements over transport and provides additional data that solidifies the discussion (e. g. higher magnetic field data). The current version of the manuscript and its main conclusions appear more solid.

I retain a small doubt about the explanation for the phonon generation at high magnetic field vs. skyrmions. There are two things that appear troublesome. 1) The authors donot consider other contributors to the high-field magnon transport such as impurities or contact transparency (STM experiments examined local spectra, while in the current manuscript magnons transport over a few micrometers across the sample). 2) The data in Fig. S5 show that the lower V_s/V_z needed to initiate alleged scattering on phonon at higher magnetic fields. In particular, it seems that if I extrapolate the threshold from the $B=5T$ to $B=2T$ (taken that the Zeeman energy will change by a factor of 2.5), the actual bias voltage that is needed to create an imbalance between "B" and "D" is the same for all fields except 1.5 T. This does not provide sufficiently strong argument towards phonon-generated scattering in the light of point 1.

It does not undermine the integrity of the main conclusions of the manuscript since the majority of the data was taken at lower magnetic fields and lower bias voltages. But I would recommend that the authors tone down discussions in the main text and remain open for other possibilities in the actual quantum Hall device they study.

I can now recommend this manuscript for publication in Nature Communications.

We thank the Reviewer for accepting our manuscript for the publication. Based on the reviewer's input, we have added the following sentences at the at the end of the first paragraph of the discussion section of the revised manuscript; **"Note that other possibilities, such as impurities or contact transparency, may affect the magnon transport at higher B."**

Reviewer #3 (Remarks to the authors)

I have read the updated version of the manuscript and the answers to the Referees' criticisms. I think that the paper in the present version is suitable for publication.

We thank the reviewer for accepting out manuscript for the publication.